# Pyrolysis of Solid Recovered Fuel Using Fixed and Fluidized Bed Reactors

**DOI:** 10.3390/molecules28237815

**Published:** 2023-11-28

**Authors:** Myeongjong Lee, Hyeongtak Ko, Seacheon Oh

**Affiliations:** Department of Environmental Engineering, Kongju National University, 1223-24 Cheonan-Daero, Seobuk, Cheonan 31080, Chungcheongnam-do, Republic of Korea; dlaudwhd77@naver.com (M.L.); kht3113@naver.com (H.K.)

**Keywords:** pyrolysis, solid recovered fuel, fixed bed reactor, fluidized bed reactor, kinetic analysis

## Abstract

Currently, most plastic waste stems from packaging materials, with a large proportion of this waste either discarded by incineration or used to derive fuel. Accordingly, there is growing interest in the use of pyrolysis to chemically recycle non-recyclable (i.e., via mechanical means) plastic waste into petrochemical feedstock. This comparative study compared pyrolysis characteristics of two types of reactors, namely fixed and fluidized bed reactors. Kinetic analysis for pyrolysis of SRF was also performed. Based on the kinetic analysis of the pyrolytic reactions using differential and integral methods applied to the TGA results, it was seen that the activation energy was lower in the initial stage of pyrolysis. This trend can be mainly attributed to the initial decomposition of PP components, which was subsequently followed by the decomposition of PE. From the kinetic analysis, the activation energy corresponding to the rate of pyrolysis reaction conversion was obtained. In conclusion, pyrolysis carried out using the fluidized bed reactor resulted in a more active decomposition of SRF. The relatively superior performance of this reactor can be attributed to the increased mass and heat transfer effects caused by fluidizing gases, which result in greater gas yields. Regarding the characteristics of liquid products generated during pyrolysis, it was seen that the hydrogen content in the liquid products obtained from the fluidized bed reactor decreased, leading to the formation of oils with higher molecular weights and higher C/H ratios, because the pyrolysis of SRF in the fluidized bed reactor progressed more rapidly than that in the fixed bed reactor.

## 1. Introduction

The past few decades have seen a continuous increase in the production and demand for plastics. This trend is attributable not only to the diverse physiochemical properties of plastics, which allow for their wide applicability, but also to their low cost. However, this surge in plastic usage has driven a corresponding increase in plastic waste. This finding is reflected in the fact that plastic constitutes more than 50% of waste produced by the average household [1]. Therefore, in light of the long-term persistence of plastic in the environment, recycling plastic waste has been highlighted as an important factor in the large-scale establishment of a circular economy and carbon neutrality [2]. In terms of the specific underlying plastic waste, approximately 40% of all plastic waste is generated from packaging materials, and approximately 60% of this waste is channeled toward energy recovery or discarded [3]. Unlike biomass, plastic waste discarded in landfills does not biodegrade; instead, this plastic undergoes decomposition that can occur for many centuries. In addition, this decomposition does not occur uneventfully, as it has been linked to the accumulation of wide-reaching pollutants in landfills [4,5]. Plastics mainly consist of LDPE (low-density polyethylene), HDPE (low-density polyethylene), PP (polypropylene), PS (polystyrene), PET (polyethylene terephthalate), and PVC (polyvinyl chloride). Among these constituents, PE, PP, and PS constitute 50–70% of most plastics [6,7]. For recycling of plastic wastes, this waste is supposed to be first sorted on the basis of its physiochemical makeup. However, considering the mixed nature of plastic waste, the processes currently used to sort it have numerous limitations; this is a serious issue, especially considering the fact that the efficiency of mechanical recycling for material reuse is dependent on the proper sorting and evaluation of plastic waste [8]. In a situation in which the sorting or separation of already heterogeneous plastic waste is not optimal, another layer of inefficiency is introduced in the already disorganized, wasteful plastic recycling industry. Additionally, mechanical recycling is typically accompanied by the deterioration of plastic properties, so alternative methods to conventional mechanical recycling must be considered, especially for ensuring the sustainable use of plastics in a circular economy [9]. Plastics consist of a wide range of hydrocarbons; this makeup translates into plastics hosting a lot of recoverable chemical energy [6,10]. Thus, chemical recycling through pyrolysis is garnering attention as an important alternative to recycling plastic waste into petrochemical feedstock. Pyrolysis is a reaction that entails the decomposition of high-molecular-weight compounds with long-chain structures into low molecular weight compounds. This process occurs under heating conditions in an oxygen-free atmosphere, with its yields being oil, non-condensable gas and solid residue [11]. Pyrolysis addresses a major drawback of traditional mechanical recycling—its inability to enable continuous recycling—by allowing for the recovery of otherwise non-recyclable waste plastics [12]. It has also been shown that the calorific value of pyrolysis-derived oil produced from plastic waste is comparable to that of conventional diesel fuel, allowing it to serve as a robust replacement. Moreover, the materials generated post-pyrolysis can be reused in existing petrochemical processes, making them an excellent alternative in the energy market [13,14,15,16,17,18].

In South Korea, a specific recycling rate is mandated for plastic packaging materials by the extended producer responsibility (EPR) system. Accordingly, this waste is either mechanically recycled or used as solid recovered fuel (SRF). In other words, waste plastics that can be mechanically sorted by type are recycled as materials, and those that cannot be mechanically sorted are converted into SRF and recycled as fuel for energy recovery. In addition, SRF is managed by the standards on the heating value, moisture content, ash content, chlorine and sulfur contents, and heavy metal contents (mercury, cadmium, lead, and arsenic) based on the Korean Waste Management Regulation. In this study, an investigation was conducted on the pyrolysis of SRF for chemical recycling to change SRF into plastics or materials that can be used as the raw materials of other products. To achieve this aim, we used both batch fixed bed and continuous fluidized bed reactors. We then comprehensively compared its yield and pyrolytic properties through the two types of reactors. Additionally, using kinetic analyses, we investigated the activation energy required for the pyrolysis of SRF.

## 2. Results and Discussion

### 2.1. Kinetic Analysis

Figure 1 shows the TG (thermogravimetric) and DTG (derivative thermogravimetric) curves, illustrating the heating rates under a nitrogen atmosphere for kinetic analysis of the pyrolysis of the SRF. The TG curve in Figure 1a shows that the decomposition temperature rose as the heating rate increased. This occurred likely due to thermal transfer lag caused by the increased heating rate. Additionally, Figure 1b’s DTG curve shows more than two peaks, confirming that the SRF used in this study was a heterogeneous mixture of different plastics. The SRF used in this study was primarily made from plastic packaging waste composed of PP and PE. Generally, PP begins to undergo pyrolysis at 400 °C, whereas PE is pyrolyzed at a higher temperature [19,20]. Therefore, the peak (i.e., at ~400 °C) in the DTG curve is likely due to the pyrolysis of PP, whereas the subsequent peak (at 450–500 °C) is likely attributable to PE. The activation energies for each conversion rate were determined by applying the differential method described in Equation (2), and these are presented in Figure 2. Figure 2b shows that the activation energy increased (from 59.9 to 116.3 kJ/mol) as the conversion rate increased to 0.35, after which it remained fairly constant (between 112.9 and 131.3 kJ/mol). This trend can be attributed to the SRF sample used in this study being a composite of plastics mostly made of PE and PP; the initial conversion rates were primarily due to PP, whereas the subsequent ones were a manifestation of the pyrolysis of PE.

Figure 3 shows the outputs of the integral methods, encapsulated in Equations (10) and (11); these outputs represent activation energies across various heating rates. The range of conversion rates for obtaining activation energy at each heating rate in Figure 3 was set based on the peaks observed in the DTG analysis in Figure 1b, and was applied in two stages. Figure 3 shows that when using the integral method, the activation energies appeared to be lower than those elucidated via the differential method (Figure 2). However, the tendency for the activation energy to be lower in the initial stages of pyrolytic conversion was also observed in results obtained via the differential method.

Table 1 shows a comparison of the activation energies for the pyrolysis of SRF based on the different kinetic methods applied. We found that in the context of the pyrolysis reactions of PP and PE, the activation energy for the pyrolysis of PP appeared to be lower than that of PE [21]. Therefore, although the different kinetic analysis methods brought forth different activation energies, the universal trend was as follows: PP was the first to decompose, followed by PE in subsequent stages. This confirmed the validity of the kinetic analysis method used, as the activation energy for the initial pyrolysis of PP is lower than that for the later stages of PE decomposition. Based on the analysis results of the kinetic analysis method applied in this study, the differential approach that uses multiple heating rates is judged to be more useful than the integral approach that uses the TGA results of the single heating rate, because it can examine the change in activation energy according to the pyrolysis conversion rate.

### 2.2. Product Analysis

#### 2.2.1. Fixed Bed Reactor

Figure 4 shows the changes in yield for each product based on the pyrolysis reaction temperature when SRF was pyrolyzed using a fixed bed reactor. As the pyrolysis reaction temperature increased, the yield of liquid products such as heavy oil and light oil decreased (Figure 4). This is likely because the gasification reactions became more active, increasing the yield of gaseous products. Additionally, even though the yield of the solid residue slightly decreased when the reaction temperature exceeded 550 °C, the temperature-dependent changes it underwent were minimal.

Figure 5 shows the characteristics of the carbon content of the liquid products obtained from GC-MS analysis of SRF pyrolysis using a fixed bed reactor. Figure 5 shows that the heavy oil mostly consisted of components with carbon numbers greater than C21, culminating in the impact of the pyrolysis reaction temperature on its composition being minimal. The heavy oil also contained components in the C5–C10 range, suggesting that more refined condensation could potentially increase the yield of light oil. However, the light oil mostly consisted of components in the C7 –C8 range. As the reaction temperature increased, there was a noticeable increase in lower-molecular-weight components, especially at 650 °C, culminating in all the components having carbon numbers of C8 or less. This is likely because the pyrolysis becomes more intense as the reaction temperature increases. However, temperatures that are too high can decrease the yield of liquid products due to the corresponding increase in the yield of gaseous products. Therefore, selecting the optimal pyrolysis reaction temperature for SRF should entail considering the yield changes for each product in Figure 4.

Figure 6 shows the gas chromatogram for liquid products at a reaction temperature of 600 °C. Heavy oil evidently has components whose heterogeneity is greater than those of light oil.

In addition, the higher heating values of the light oil and heavy oil generated from the fixed bed reactor were found to be 9502.0 to 11,395.5 kcal/kg and 6868.2 to 8164.4 kcal/kg, respectively, indicating that the higher heating value of the light oil was higher than that of the heavy oil. For the gas products released through the pyrolysis of waste plastics, they are hydrocarbon compounds with carbon numbers of C4 or less according to several studies [22,23,24]. In the case of the characteristics of the solid residues obtained from the pyrolysis of the fixed bed reactor in this study, the higher heating value ranged from 4570.8 to 5421.2 kcal/kg, as shown in Table 2, and the elemental analysis results confirmed that the content of residual carbon was not low.

#### 2.2.2. Fluidized Bed Reactor

Figure 7 shows the changes in yield for each product based on the pyrolysis reaction temperature and the fluidized gas flow rate when SRF was pyrolyzed using a fluidized bed reactor.

Figure 7 shows that compared to results obtained via a fixed bed reactor, as the reaction temperature increased, the yield of gaseous products significantly increased, substantially decreasing the yield of heavy oil. Fluidized bed reactions are known to enhance mass and heat transfer compared to fixed bed reactions in the pyrolysis of waste plastics [25]. This trend was further confirmed by the increase in gaseous products as the fluidizing gas velocity increased. Additionally, compared to the fixed bed reactor, the fluidized reactor produced a greater yield of gaseous products; this is because fluidized reactors inherently facilitate decomposition (i.e., into smaller molecular weight components) that is more intensive than that seen in fixed bed reactors. However, in the case of light oil, while a slight decrease was observed as the reaction temperature increased in fixed bed reactions, an increase was noted in fluidized bed reactions. Therefore, as previously mentioned, we confirmed that SRF decomposed into lower-molecular-weight components more readily in fluidized bed reactions compared to fixed bed reactions. For the solid residues, temperature-driven changes were nearly negligible, similar to those in fixed bed reactions.

Figure 8 shows the characteristics of the carbon content of the liquid products obtained from GC-MS analysis of SRF pyrolysis using a fluidized bed reactor. Figure 8 suggests that, similar to fixed bed reactions, components with a carbon number of C21 and above were most prevalent in the heavy oil. However, the proportion of these high-carbon components was higher in fluidized bed reactions, especially at 650 °C, where they were produced in substantial amounts. This could be due to the pyrolysis of SRF in fluidized bed reactors progressing more rapidly than that in fixed bed reactors. As a result, the hydrogen content in the liquid products decreases, leading to the formation of oils with higher molecular weights, and higher C/H ratios. Additionally, in the case of light oil, components ranging from C7–C8 predominated, similar to what we saw in fixed bed reactions. We observed that lower-molecular-weight components increased as the reaction temperature and fluidizing gas velocity increased.

Figure 9 shows the gas chromatograms of heavy and light oil obtained at a reaction temperature of 600 °C and a fluidizing gas flow rate of 1.0 L/min. This trend shows that the composition of heavy oil was more diverse than that of light oil, consistent with the results of the fixed bed reactor.

In addition, the higher heating values of the light oil and heavy oil generated from the fluidized bed reactor were found to range from 9411.5 to 10,271.4 kcal/kg and 6702.4 to 7736.6 kcal/kg, respectively, and the higher heating value of the light oil was also higher than that of the heavy oil, as in the case of the fixed bed reactor. Table 3 shows the characteristics of the solid residues obtained from the pyrolysis of the fluidized bed reactor in this study. It was found that the higher heating value ranged from 4853.6 to 5836.2 kcal/kg, and that the content of residual carbon was not low as in the case of the fixed bed reactor.

## 3. Materials and Methods

### 3.1. Materials

In this study, SRF produced from plastic packaging waste, such as PP and PE, was mainly used as a pyrolysis sample. The characteristics of the SRF used in this study are shown in Table 4. The composition of SRF was then analyzed using an elemental analyzer (EA) (Flash 2000, UK). As can be seen from the table, the main components of the SRF used in this study were hydrogen and carbon as it was mostly composed of plastics, and it is judged that the content of PET was not low considering the oxygen content. The presence of nitrogen appeared to be due to the ABS (acrylonitrile butadiene styrene) content. The fact that most of the packaging used was made of plastic is reflected in the high amount of volatile matter (i.e., 90.6%) that was detected, while the fixed carbon and ash were recorded at 6.07% and 3.33%, respectively.

### 3.2. Experimental Methods

The schematic diagrams of the fixed bed and fluidized bed reactors used in this study’s pyrolysis experiments are shown in Figure 10. The fixed bed reactor (Figure 10a) consisted of a quartz tube with a diameter and length of 60 and 550 mm, respectively. The pyrolysis products were collected through a two-stage cooler using cooling water and dry ice. For the fixed bed pyrolysis experiment, an initial 15 g sample was loaded into a ceramic boat and inserted into the reactor. Heating, which was performed at a rate of 10 °C/min, was controlled using a PID controller until reaction temperatures of 550, 600, and 650 °C were reached. The reaction time was maintained for 30 min after reaching the predetermined temperature for the sufficient pyrolysis of SRF. The resultant liquid recovered from the cooler and solid residues were then analyzed. The yield of gaseous products was calculated based on the initial sample weight and the amounts of liquid and solid residues. The fluidized bed reactor (Figure 10b) was designed to allow continuous sample injection, unlike the fixed bed reactor. It consisted of a stainless-steel tube, with a lower section that had a diameter and length of 62 and 370 mm, respectively. Its upper section had a diameter and length of 26 and 730 mm, respectively. As in the use of the fixed bed reactor, the pyrolysis products were collected through a two-stage cooler using cooling water and dry ice. In addition, 50 g of sand with a diameter of 125–180 µm was used as the fluidizing medium for the fluidized bed reaction. The fluidized bed pyrolysis experiments were conducted at nitrogen flow rates ranging from 0.75–1.5 L/min, allowing for optimal fluidization. These flow rates were adjusted at an interval of 0.25 L/min. Upon reaching the set reaction temperature, samples were continuously injected at a rate of 0.5–0.7 g/min for 1 h. Like in the case of the fixed bed reactor, pyrolysis was also set to occur at 550, 600, and 650 °C, and these temperatures were controlled using a PID controller. Finally, yields of the liquid products were analyzed once the resultant products were collected from the cooler. The amount of solid residue was analyzed by weighing the solid material remaining in the reactor and the initial sand used as the fluidizing medium after the experiment. The yield of gaseous products was calculated similarly for the fixed bed experiments, based on the initial SRF sample weight and the amounts of liquid and solid residues. In addition, the pyrolysis experiments performed using the fixed and fluidized bed reactors were repeated five times for the reliability of the experiment results.

The composition of the liquid products recovered through the fixed and fluidized bed reactors was analyzed using gas chromatography mass spectrometry (GC-MS) (GCMS-QP2010 Ultra, Dong-il SHIMADZU, Japan). The conditions under which the GC-MS was operated are presented in Table 5. Additionally, for the kinetic analysis of the pyrolysis reactions, thermal mass changes were analyzed using thermogravimetric analysis (TGA) (Pyris 1 TGA, USA) at heating rates of 20, 30, and 40 °C/min under a nitrogen atmosphere. The higher heating value of all samples was determined using a bomb calorimeter (Parr Instrument Co., Model 1672, Moline, IL, USA).

### 3.3. Kinetic Analysis

The reaction rate equation for the conversion rate of pyrolysis can be represented via an Arrhenius Equation (1). In the case of kinetic analysis based on TGA, different results are generally derived depending on the analysis method [26]. Therefore, in this study, the differential approach that comprehensively uses the TGA results of multiple heating rates and the integral approach that individually uses the TGA results of the single heating rate were utilized to examine the validity of the kinetic analysis results. Differential and integral approaches are widely used in the kinetic analysis of pyrolytic reactions using TGA results [26].
(1)$$\frac{d\alpha }{\mathrm{dt}}={\mathrm{Ae}}^{-\frac{E}{\mathrm{RT}}}{\left(1-\alpha \right)}^{n}$$
where A: pre-exponential factor (min^−1^)

E: apparent activation energy (kJ/mol)

n: apparent order of reaction

R: gas constant (8.3136 J/mol·K)

T: absolute temperature (K)

t: time (min)

α: degree of conversion.

#### 3.3.1. Differential Method

This method utilizes the following logarithmic differential equation derived from Equation (1).
(2)$$\ln\left(\frac{d\alpha }{\mathrm{dt}}\right)=\ln\left\{A{\left(1-\alpha \right)}^{n}\right\}-\frac{E}{\mathrm{RT}}$$

The first term on the right side of Equation (2) is constant for a fixed conversion rate, α. Therefore, when ln(da/dt) and 1/T are plotted at heating rates of 20, 30, and 40 °C/min, which were used in TGA of this study, for each fixed conversion rate, activation energy E can be obtained from the slope.

#### 3.3.2. Integral Method

By rearranging Equation (1) using the linear heating rate β (K/min), Equation (3) can be derived; Equation (3) can then be further rearranged to yield Equation (4).
(3)$$\frac{d\alpha }{\mathrm{dT}}=\frac{A}{\beta }e^{-\frac{E}{\mathrm{RT}}}{\left(1-\alpha \right)}^{n}$$
(4)$$\frac{d\alpha }{{\left(1-\alpha \right)}^{n}}=\frac{A}{\beta }e^{-\frac{E}{\mathrm{RT}}}\mathrm{dT}$$

The application of the integral approximation method [27] to the right side of Equation (4) yields the following [28].
(5)$$\int_{0}^{\alpha }\frac{d\alpha }{{\left(1-\alpha \right)}^{n}}=\frac{1-{\left(1-\alpha \right)}^{1-n}}{1-n}\;\;\;n\neq 1$$
(6)     =−ln(1−α)   n=1
and
(7)$$A/\beta \int_{0}^{T}e^{-E/\mathrm{RT}}\mathrm{dT}\;\approx \;\frac{{\mathrm{ART}}^{2}}{\beta E}\left(1-\frac{2\mathrm{RT}}{E}\right)e^{-E/\mathrm{RT}}$$

Additionally, taking the logarithm of Equations (5)–(7) yields Equations (8) and (9).
(8)$$\ln\left\{\frac{1-{\left(1-\alpha \right)}^{1-n}}{T^{2}\left(1-n\right)}\right\}=\ln\frac{\mathrm{AR}}{\beta E}\left(1-\frac{2\mathrm{RT}}{E}\right)-\frac{E}{\mathrm{RT}}\;\;\;n\neq 1$$
(9)$$\ln\left\{\frac{-\ln{\left(1-\alpha \right)}^{1-n}}{T^{2}}\right\}=\ln\frac{\mathrm{AR}}{\beta E}\left(1-\frac{2\mathrm{RT}}{E}\right)-\frac{E}{\mathrm{RT}}\;\;\;n=1$$

Therefore, after assuming each reaction order, the activation energy values can be determined from the slopes of the plotted relations.
(10)$$Y=-\ln\left\{\frac{1-{\left(1-\alpha \right)}^{1-n}}{T^{2}\left(1-n\right)}\right\}\mathrm{vs}\frac{1}{T}\;\;\;n\neq 1$$
(11)$$Y=-\ln\left\{-\frac{\ln\left(1-\alpha \right)}{T^{2}}\right\}\mathrm{vs}\frac{1}{T}\;\;\;n=1$$

Using Equations (10) and (11), the reaction orders n can be determined for the cases that best fit a straight line, and the activation energies can be obtained from the slopes of those lines.

## 4. Conclusions

In this study, we compared the pyrolytic properties of the SRF made from plastic waste using two reactors, namely, fixed bed and fluidized bed reactors, and kinetic analysis for pyrolysis of SRF was performed. Based on the kinetic analysis of the pyrolytic reactions using differential and integral methods applied to the TGA results, we found that the activation energy was lower in the initial stage of pyrolysis. This trend can be mainly attributed to the initial decomposition of PP components, which was subsequently followed by the decomposition of PE. Based on the analysis results of the kinetic analysis method applied in this study, the differential approach that uses multiple heating rates is judged to be more useful than the integral approach that uses the TGA results of the single heating rate because it can examine the change in activation energy according to the pyrolysis conversion rate. Using a fixed bed reactor, we found that as the reaction temperature increased, gasification became more vigorous, resulting in an increase and decrease in the yields of the gas and liquid, respectively. Via the fluidized bed reactor, we confirmed that, similar to the case of the fixed bed reactor, the gas yield increased as the reaction temperature increased and also rose as the flow rate of fluidizing gas increased. Overall, fluidized bed reactions resulted in a more active decomposition of SRF. The relatively superior performance of this reactor can be attributed to the increased mass and heat transfer effects caused by fluidizing gases, which result in greater gas yields. Regarding the characteristics of liquid products generated during pyrolysis, it was seen that the hydrogen content in the liquid products obtained from the fluidized bed reactor decreased, leading to the formation of oils with higher molecular weights and higher C/H ratios, because the pyrolysis of SRF in fluidized bed reactors progresses more rapidly than that in fixed bed reactors.

## Figures and Tables

**Figure 1 molecules-28-07815-f001:**
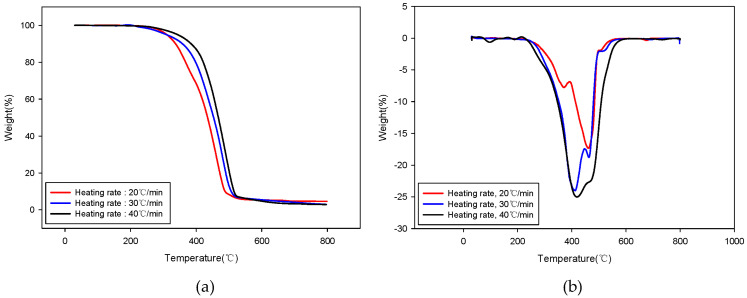
TG (**a**) and DTG (**b**) curves of SRF across various heating rates in pure nitrogen.

**Figure 2 molecules-28-07815-f002:**
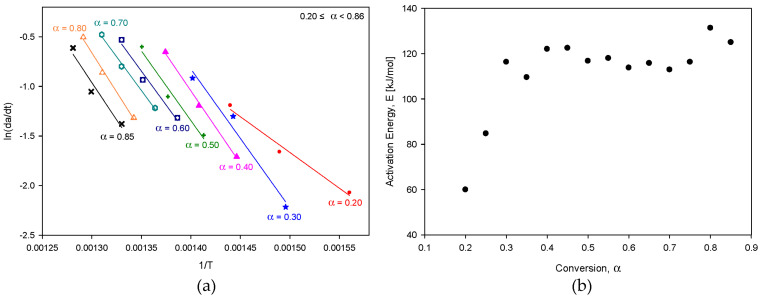
Application of differential method (**a**) and activation energies (**b**) over a range of conversion.

**Figure 3 molecules-28-07815-f003:**
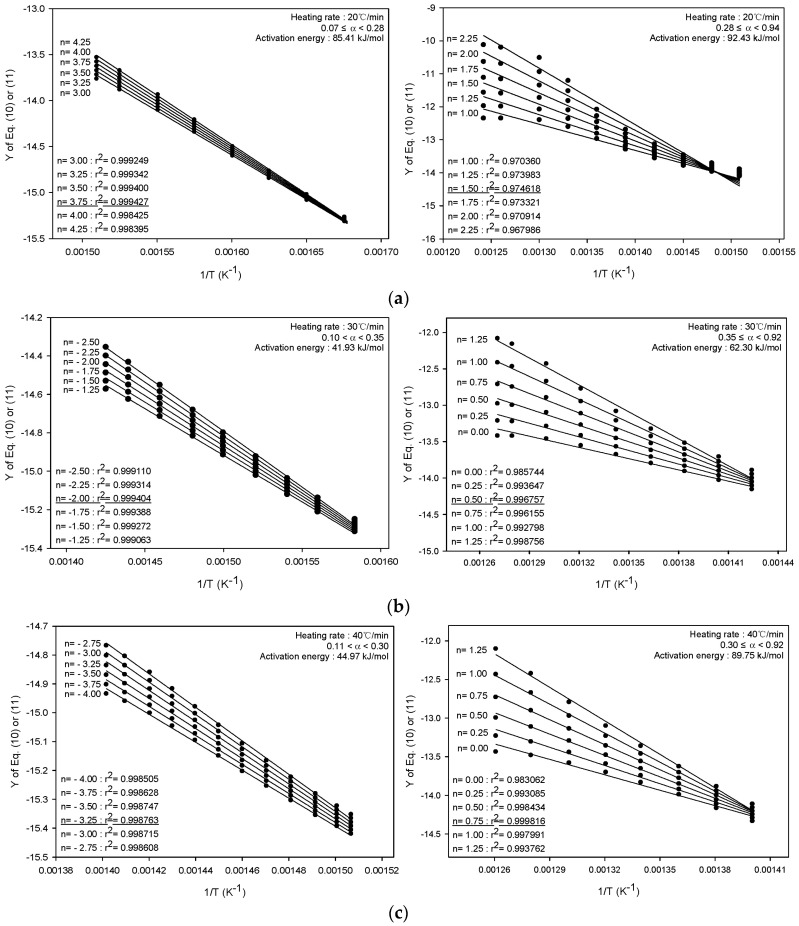
Activation energies at various heating rates of 20 °C/min (**a**), 30 °C/min (**b**), and 40 °C/min (**c**), as determined using the integral method.

**Figure 4 molecules-28-07815-f004:**
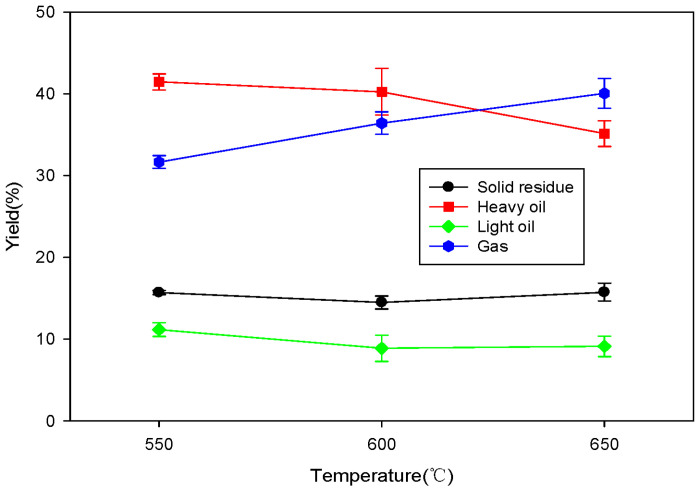
Product yields of SRF pyrolysis using fixed bed reactor across different temperatures.

**Figure 5 molecules-28-07815-f005:**
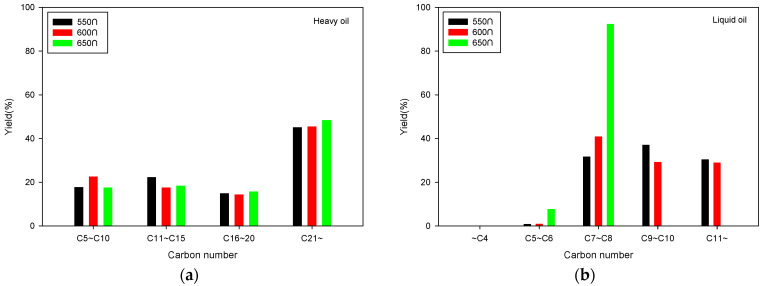
Compounds according to the carbon number of heavy (**a**) and light oil (**b**) obtained from fixed bed reactor.

**Figure 6 molecules-28-07815-f006:**
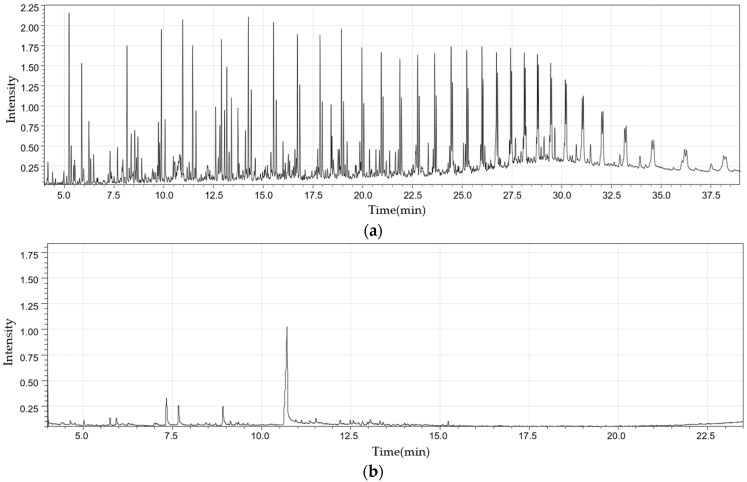
Gas chromatogram of heavy (**a**) and light oil (**b**) obtained from a fixed bed reactor at a reaction temperature of 600 °C.

**Figure 7 molecules-28-07815-f007:**
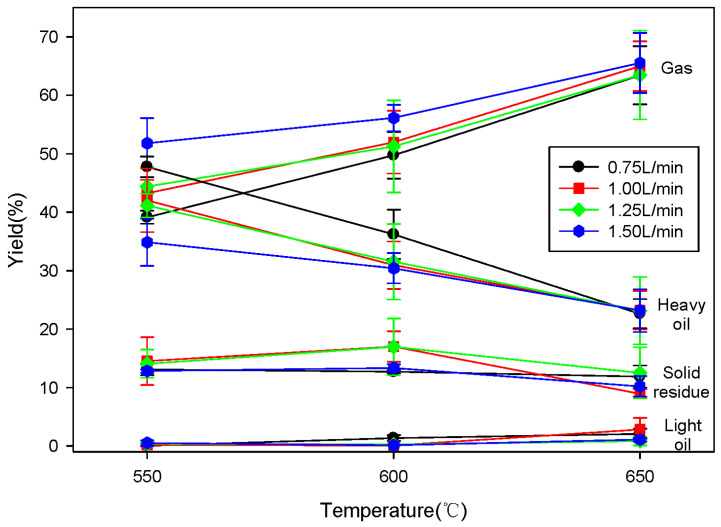
Product yields of SRF pyrolysis using fluidized bed reactor according to temperatures and fluidized gas flow rate.

**Figure 8 molecules-28-07815-f008:**
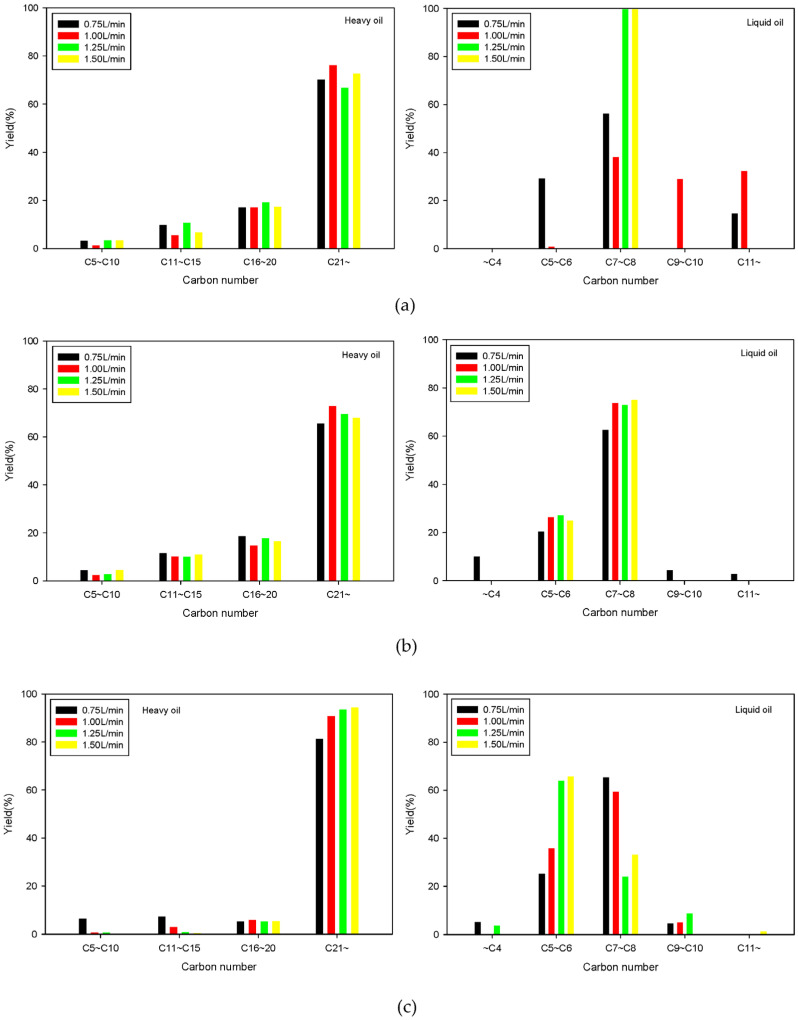
Compounds according to the carbon number of heavy and light oil obtained from fluidized bed reactor at reaction temperatures of 550 (**a**), 600 (**b**), and 650 °C (**c**).

**Figure 9 molecules-28-07815-f009:**
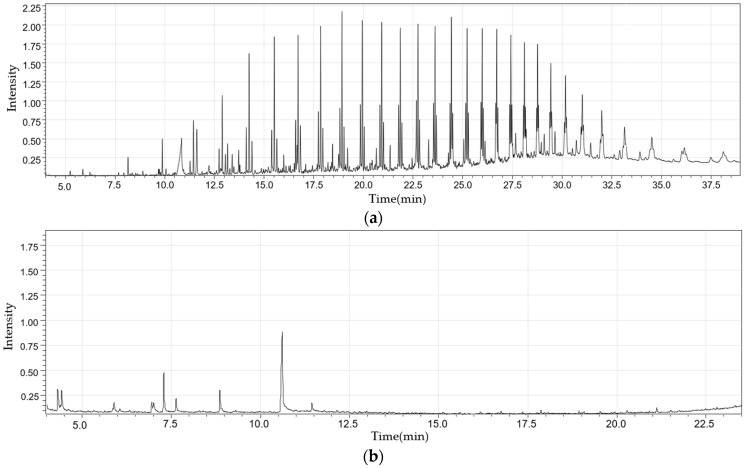
Gas chromatogram of heavy (**a**) and light oil (**b**) obtained from a fluidized bed reactor at a reaction temperature of 600 °C and a gas flow rate of 1.0 L/min.

**Figure 10 molecules-28-07815-f010:**
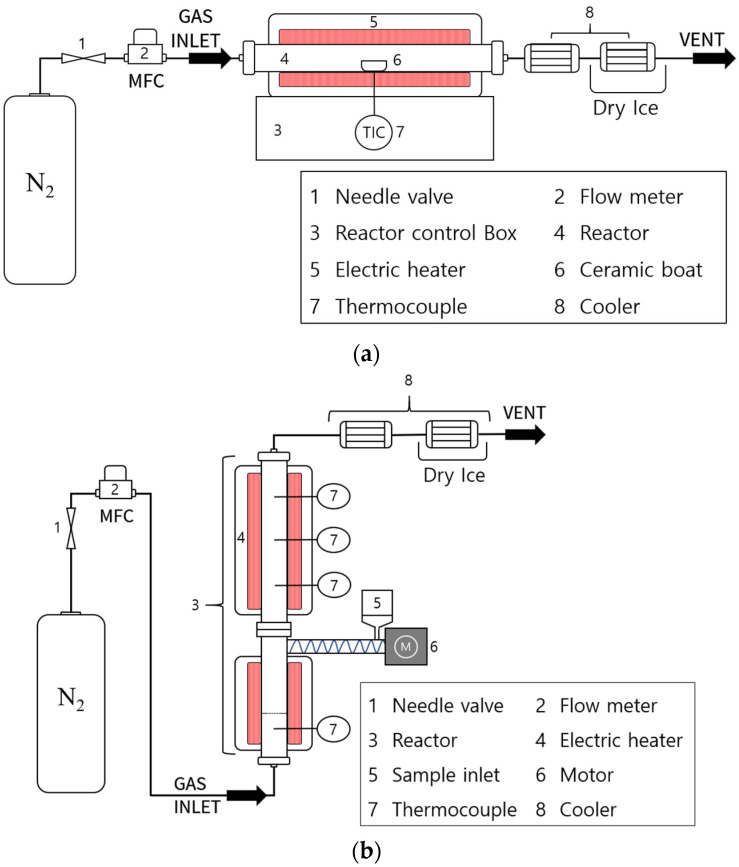
Schematic diagram of fixed (**a**) and fluidized (**b**) bed reactors used in this study.

**Table 1 molecules-28-07815-t001:** Activation energy of pyrolysis of SRF used in this work.

Differential Method		Integral Method		
Conversion (α)	Activation Energy (kJ/mol)	Heating Rate	Conversion (α)	Activation Energy (kJ/mol)
0.20 ≤ a < 0.40	59.94~116.32	20 °C/min	0.07 ≤ a < 0.28	85.41
			0.28 ≤ a < 0.94	92.43
		30 °C/min	0.10 ≤ a < 0.35	41.93
0.40 ≤ a < 0.85	112.47~131.32		0.35 ≤ a < 0.92	62.30
		40 °C/min	0.11 ≤ a < 0.30	44.97
			0.30 ≤ a < 0.92	89.75

**Table 2 molecules-28-07815-t002:** Properties of solid residues obtained from fixed bed reactor.

Elements (wt%)	C	42.17–46.39
	H	1.41–2.39
	N	0.90–1.16
	O	4.18–7.63
	S	0
	Other	45.44–51.46
Higher heating (kcal/kg)	4570.8–5421.2	

**Table 3 molecules-28-07815-t003:** Properties of solid residues obtained from fluidized bed reactor.

Elements (wt%)	C	40.40–44.16
	H	1.95–3.31
	N	0.94–1.34
	O	6.41–9.75
	S	0
	Other	45.87–46.53
Higher heating (kcal/kg)	4853.6–5836.6	

**Table 4 molecules-28-07815-t004:** Properties of SRF used in this work.

Elements (wt%)	C	63.1
	H	9.8
	N	0.6
	O	16.4
	S	0
	Other	10.1
Volatile (%)	90.60	
Fixed carbon (%)	6.07	
Ash (%) Higher heating value (kcal/kg)	3.33 8871.7	

**Table 5 molecules-28-07815-t005:** Conditions under which the gas chromatography mass spectrometry (GC-MS) was used.

Item	Conditions
Column over temp	40.0 °C
Injection temp	250.0 °C
Injection mode	Split
Flow control mode	Linear Velocity
Pressure	60.1 kPa
Total flow	234.1 mL/min
Column flow (He)	1.15 mL/min
Linear velocity	38.6 cm/s
Purge flow	3.0 mL/min
Split ratio	200.0

## Data Availability

The data presented in this study are available on request from the corresponding author. The data are not publicly available due to privacy.

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
