# Peer review of "Pyrolysis of Solid Recovered Fuel Using Fixed and Fluidized Bed Reactors"

_molecules, 2023, doi:10.3390/molecules28237815_

Round 1
Reviewer 1 Report
Comments and Suggestions for Authors
This manuscript describes the pyrolysis of a plastic packaging waste mainly composed of PP and PE in both fixed bed and fluidized bed reactors. The effects of pyrolysis temperature and fluidized gas flow rate (in fluidized bed) are compared. The experimental workload is relatively large, the data analysis is more detailed, and the results have certain reference value, but the innovation is general. Based on my comprehensive consideration, I would recommend that the manuscript must be majorly revised for publication.
Specific comments:
1. Why is plastic packaging waste specifically called solid recyclable fuel?
Two methods, both the differential and integral approaches, were used to analyze the TGA pyrolysis kinetics and calculate the activation energy. The difference between the two approaches is very significant. What is the necessity or importance of doing so? It is recommended to choose one of them.
2. The analysis of pyrolysis products is insufficient. In addition to the composition of the liquid oil as part of the pyrolysis product, the composition of the gas and solid residues should also be identified.
3. The combustion thermal properties of gas and liquid pyrolysis products need to be measured.
4. The academic quality of abstracts and conclusions needs to be improved, and quantitative conclusions expressed in terms of data are not presented at all.
Author Response
Dear reviewer,
Thank you very much for your precious review and comments. I tried to make most of the editing to its best. I have included the discussions and corrections for your comments. I would be glad to answer and make those corrections. I am sure it will help make my paper a good one.

Reviewer 2 Report
Comments and Suggestions for Authors
This paper investigated the preparation of pyrolytic products from SRF using fixed and fluidized bed reactors and their characteristic analyses. Based on the following comments, this manuscript should be revised prior to the publication.
1. In general, SRF is often used an auxiliary fuel in the industrial boilers. Therefore, it is an interesting study. In the introduction, the authors should summarize the status of SRF use and its regulatory specifications/requirements in South Korea.
2. The authors should list the calorific values of the SRF feedstock and its resulting pyrolytic products, especially in liquid products. In addition, what are the contents of inorganic elements in ash?
3. In the experimental pyrolysis conditions, what are the heating rate and residence time in the fixed bed and fluidized bed reactors?
4. Based on the results in the TGA, the authors should elucidate the process conditions adopted in this work.
5. According to my experience in the pyrolysis, the data on the yields of the resulting products were not often consistent. Thus, the repeatability of the data should be stated in detail.
Author Response

(The authors gave the same response as above.)

Reviewer 3 Report
Comments and Suggestions for Authors
All comments are in text file.

The manuscript does not comply with the norms of traditional English. As an example, here are a number of expressions:
"it first has to be sorted based on its physiochemical make";
"sorting processes used to scan for contaminants";
"Plastics ... are a mosaic of hydrocarbon compounds";
etc.
Author Response

(The authors gave the same response as above.)

Round 2
Reviewer 3 Report
Comments and Suggestions for Authors
Equation 5 is still wrong. It is clear now, that equation was taken from [28] (eq. 13) and it is different in that source.
Author Response
Dear reviewer,
Thank you very much for your precious review and comment. The manuscript has been revised as per the content pointed out by the reviewer. I am sure your review will help make my paper a good one.
